# SpaceART SpaceWire Sniffer for Link Monitoring: A Complete Communication Analysis in a Time-Constrained Test Scenario [note 1]

**DOI:** 10.3390/s23031580

**Published:** 2023-02-01

**Authors:** Roberto Ciardi, Simone Vagaggini, Antonino Marino, Pietro Nannipieri, Luca Fanucci

**Affiliations:** 1Department of Information Engineering, University of Pisa, 56122 Pisa, PI, Italy; 2Space Division, IngeniArs S.r.l., 56121 Pisa, PI, Italy

**Keywords:** SpaceWire, EGSE, OBDH, SpaceART, link analyser, sniffer, spacecraft, test equipment, validation

## Abstract

The on-board communication standard adopted in current generation space missions of the European Space Agency, and many other agencies as well, is SpaceWire (SpW). In a SpW network, data are exchanged as well-formed packets, whose structure offers low packet overhead and allows developers to tailor their implementation for SpW applications. The development of SpW-compliant devices requires a specific set of test instruments, namely Electrical Ground Support Equipment (EGSE), to verify the correct functionality of SpW units under test. An example of a SpW EGSE is the SpaceART EGSE Emulator, an EGSE for generation and processing of SpW packets for a time-constrained use-case scenario. This EGSE has been developed to address a mission-related SpW communication between a device and an Instrument Control Unit (ICU). It has allowed the generation and processing of specific SpW packets, which cannot be provided by the mostly used general-purpose SpW EGSEs. In this scope, the SpaceART SpW Sniffer, a SpW link analyser for unobtrusive monitoring of SpW link, has been employed to run a comprehensive set of tests and provide further information on the considered scenario. The SpaceART EGSE Emulator-ICU communication has been thoroughly tested through the SpaceART SpW Sniffer, unobtrusively analysing the exchanged data and allowing to assess the compliance with the defined time constraints from an external point of view. The use of the Sniffer has been crucial for testing the on-board communication, providing important support for the success of the mission employing the tested ICU.

## 1. Introduction

SpaceWire (SpW) is the communication standard for Spacecrafts on-board electronics, adopted in many space missions of the European Space Agency (ESA) (e.g., Gaia, ExoMars, Sentinel, etc.), and in missions of many other space agencies as well (e.g., BepiColombo, ASTRO-H, SWIFT, James Webb Space Telescope, etc.) [1,2]. SpW is based on point-to-point links in which data are exchanged in well-formatted SpW packets, as defined in the SpW Standard [3]. The SpW packet structure allows to tailor the implementation of SpW communication for specific applications, allowing on-board devices to automatically communicate during the lifecycle of a space mission.

To assess the correct functionalities of SpW-based electronics, and other spacecraft electrical functions as well, specific instrumentation is needed. Such instrumentation is part of the Electrical Ground Support Equipment (EGSE): a set of tools belonging to the ground segment of a space mission, hence not intended to be launched on the spacecraft [4]. SpW EGSEs are critical to verify the general functionalities of SpW-based devices (e.g., SpW connection), as well as the mission-related functionalities (e.g., specific SpW packets incapsulation, timing, etc.). In particular, Ref. [5] describes the implementation of an EGSE for SpW data generation and processing, adopted for testing a specific time-constrained scenario.

Among EGSE, link monitoring analysers (i.e., sniffers) are essential tools to verify data traffic generated over SpW links. A link analyser provides unobtrusive link monitoring, allowing to capture data traffic over a SpW link, without affecting the communication. The SpaceART SpaceWire Sniffer is a SpW Link Analyser, which allows users to analyse SpW traffic between two nodes, identifying the exchanged SpW packets and their contents [6,7].

The SpaceART EGSE Emulator presented in [5] has been developed for emulating a specific on-board communication to test an Instrument Control Unit (ICU). The EGSE is responsible for transmitting well-formatted telemetry packets, following specific timing constraints. For this reason, a set of tests has been defined for validating both the correct transmission of data and the correct timing.

This paper presents the use of the SpaceART SpaceWire Sniffer, applied to the time-constrained test scenario described in [5]. The Sniffer is used to capture and inspect the data flowing through the SpW connections involved and verify compliance with the time constraints, by analysing the SpW traffic. The objective of the paper is to show how multiple EGSEs can be combined together to run a detailed set of tests for a Device Under Test (DUT), identifying any possible SpW communication bug in the Assembly Integration and Test phase (AIT) of a space mission. The use of a link analyser has allowed evaluating the communication from an external point of view, individuating misbehaviours in the SpW communication and acting to correct those in the development phase. Overall, this process has allowed effective verification of the DUT, stimulating any possible corner case, thanks to the transmission of specific SpW packets and the inspection of the SpW communication as well. The remainder of the paper is organized as follows:Section 2 provides information about the SpaceWire Standard, the state-of-the-art of SpW EGSEs, with particular attention to SpW Link Analysers and to the SpaceART EGSE Emulator presented in [5].Section 3 gives an overview of the SpaceART SpaceWire Sniffer and how it can be used to monitor exchanged data in a SpW communication.Section 4 delves into the test ran using the SpaceART SpaceWire Sniffer and the SpaceArt EGSE Emulator over a specific test-scenario.Finally, Section 5 and Section 6 draw the results of the described tests and conclusion of the work.

## 2. Related Work

### 2.1. SpaceWire Standard

SpaceWire is the state-of-the-art concerning the spacecraft on board communication links. SpW standard enables the communication between all the payload instrumentations, on-board computers, peripherals, and high data-rate sensors. SpW links are full-duplex bidirectional serial links, which can operate at data-rates from 2 Mbps up to 400 Mbps. As explained in the standard definition [3], the SpW standard is a well-layered protocol, comprising: Physical Layer, Signal Layer, Character Layer, Exchange Layer, and Packet layer.

Figure 1 shows the standard structure of a SpW packet. It has no limit on its size and contains three main fields: the destination address, i.e., the identity of the destination node or, alternatively, a path through a SpW network, the cargo, i.e., the data to be transferred, and the End Of Packet (EOP).

### 2.2. State-of-the-Art of SpW EGSE and Link Analysers

Since EGSEs play a key role during the integration and validation phases of the spacecraft life-cycle and given the large adoption of SpW protocol, some test equipment capable of supporting this standard have been designed, such as:STAR-Dundee **SpaceWire EGSE and Device Simulator Mk2**, which comprises two SpW interfaces and makes use of USB3.0 as a Host-PC interface [8,9];4Links **Diagnostic SpaceWire Interface (DSI)**, which comprises eight SpW interfaces [10];IngeniArs **SpaceART**, a general purpose SpW EGSE, including four SpW interfaces and Ethernet/PCIe interface to Host-PC [11,12,13];Some SpW/SpaceFibre (SpFi) EGSE solution based on PXI industry standard have also been presented in [14,15].

These systems are conceived to be used as general-purpose platforms for SpW interface testing, not allowing the testing of all corner cases defined for each specific space mission. In general, they are used as a SpW gateway, to filter data traffic and provide SpW packets to/from DUT to/from other higher-level EGSEs. On the other side the SpaceART EGSE Emulator, presented in [5], has been developed to cover a specific test scenario to generate and process definite SpW packets.

Regarding SpW Link analysis different solution have been presented. Ref [16] defines a simulator for SpW networks traffic, to provide a high-level analysis of data traffic. A more low level solution is provided by SpW Link Analysers, i.e., Sniffers. On the market, two main solutions are currently present:STAR-Dundee **SpaceWire Link Analyser Mk3** [17];IngeniArs **SpaceART SpaceWire Sniffer** [6,18];

Both instruments enable the end-user to unobtrusively monitor the traffic on a SpW link, allowing to capture and display portions of data exchanged between two SpW nodes. They are meant to show the content of the SpW packets, allowing the user to investigate SpW traffic and test SpW communication with the deserved accuracy. The details about the SpaceART SpaceWire Sniffer are described in Section 3.

### 2.3. SpaceART EGSE Emulator

The SpaceART EGSE Emulator is a customization of the SpaceART EGSE, presented in [5]. It has been used to test the interface of an ICU communicating simultaneously with three devices—referred to as Devices—on three distinct SpW links. This communication is based on Telemetry (TM) and Telecommand (TC) packets, a type of packet usually exchanged over space links [19]. The SpaceART EGSE Emulator was developed to emulate the three devices and each related SpW interface, communicating with the ICU (Figure 2), transmitting/receiving TM/TC packets.

The test scenario presented in [5] was thought not only to test the content of the SpW packet transmitted/received by the ICU, but also to test the precise time constraints to comply with, in the communication. In particular, each emulated device receives a periodical synchronization signal (SYNC) from the ICU, with a user-defined period, and replies to it, transmitting one or two TM packets (depending on the configuration of the device). On the other side, the ICU receives information about the SYNC period by the devices (inside the TM packets) and can send a TC packet to set a new configuration for the next synchronization period. All the packets must be transmitted inside a SYNC period, with the TM1 packet, sent by the devices, that must be transmitted within 3 ms from the reception of a SYNC signal, and the TM2 packet (i.e., the longer packet) that must be transmitted at least 700 us after the reception of the SYNC signal, with no end-time limit other than the SYNC period.

The behaviour of the devices is emulated by the SpaceART EGSE Emulator, receiving the SYNC signals from the ICU and sending back the properly formatted TM packets, to test the ICU operation. The time constraints for ICU-Device communication are resumed in Figure 3.

Figure 3 shows that the device transmits a TM1 packet after the reception of a SYNC signal, and an eventual TM2 packet before the end of the SYNC period (4 ms to 120 s). The ICU transmits zero or one TC packet per period.

An overview about the TM packets that are generated and sent in specific timeslots, by the SpaceART EGSE Emulator [5], follows. Ref. [5] also reports further details about the SpW characters composing each packet.

#### 2.3.1. TM1 Packet Format

TM1 packet is a 141 bytes SpW packet sent by a device as a response to the reception of a SYNC signal. The SpaceART EGSE Emulator is responsible for transmitting the TM1 packet immediately after the reception of a SYNC signal, and within 3 ms after the reception of the SYNC.

The TM1 packet carries information about the device operation in the scope of the space mission, which details are not discussed in this scope. It has logical address 0xDE and includes information about the previous SYNC period, as the number of TC packet previously received. It is also used to notify eventual errors on the received TC packets. The SpaceART EGSE Emulator allows the injection of these errors into the ICU, in order to test the ICU functionalities [5]. In particular, the SpaceART EGSE Emulator allows the User to configure the TM1 packet to be sent to the ICU. The errors are injected manually by setting two fields of the TM1 packet, named “TC Error” and “Device Error”. The set of errors that can be injected include:Errors about SpW communication (e.g., SpW parity error, SpW escape sequence error, etc.);Errors about TC packet reception (e.g., TC length error, No TC received in the SYNC period, more than one TC received in the SYNC period, etc.). Note that the N-^th^ TM1 packet contains errors about the (N−1)^th^ TC packet;Errors about device operations.

These errors can all be set and emulated by the SpaceART EGSE Emulator, to stimulate and test ICU reactions.

#### 2.3.2. TM2 Packet Format

The TM2 packet is an M bytes SpW packet, with M defined by the user in a range from 65,535 to 262,140 bytes. This packet is much simpler than the TM1 packet, carrying only dummy data except for a 4 bytes counter and a 2 bytes checksum. This packet must be transmitted at least after 700 µs from SYNC reception and within the end of the SYNC period.

## 3. SpaceART SpaceWire Sniffer

The SpaceART SpaceWire Sniffer is a link analyser part of SpaceART product family. SpaceART (Figure 4) is a complete testing solution for high-speed links in space applications [12,13]. It can operate as a standalone general-purpose SpW EGSE, generating and consuming SpW packets in real-time, or it can be customized to provide specific behaviour, as in [5], where it is used to generate and process data to emulate a distinct scenario. The SpW analysis of the SpaceART has been accurately verified through the VIP solution described in [20].

The Sniffer is specifically designed to support the test and debug phases of SpW-based systems allowing the analysis of SpW traffic between two nodes at SpW character level, without interacting with the communication.

It is designed for those applications which require analysis of long SpW communications (∼hours) and advanced trigger conditions, to facilitate the acquisition of portions of interest. In particular, the Sniffer comprises 2 SpW ports—with a traffic speed up to 200 Mbps—compliant with the SpW standard, and a trace memory for each SpW interface. The presence of four SMA connectors allows the use of external synchronization mechanisms that can be useful, for example, to synchronize the time between different SpW units or to inject link disconnections.

The Sniffer provides advanced trigger conditions: hence it allows to define a specific event that triggers the process of data saving. Some trigger events are:A specific number of packets transmitted on the link;The transmission of one or more control characters (e.g., FCT, NULL, ESC);The transmission of a specific SpW character;The transmission of a set of consecutive SpW characters part of a SpW packet;

All the trigger events can also be chained, to provide an accurate examination of the communication. When the trigger event occurs, the data traffic flowing on each side of the SpW connection is stored by the Sniffer and sent to a Host-PC, connected through a 1 Gigabit-Ethernet interface. On the Host-PC, an easy-to-use Graphical User Interface (GUI) provides features to define the described trigger conditions and to efficiently analyse the stored data. In particular, the SpW data are saved in a database file (i.e., a storable and portable file) and can be visualized and navigated both at packet and character level.

Some screenshots of the SpaceART Sniffer GUI shows the data captured in a communication test between the SpaceART EGSE and a SpaceWire DUT. Figure 5 shows a portion of communication between the two devices, where the exchanged SpW packet are arranged in a table, in the centre of the figure. The left columns of the table, below the “SpaceART EGSE” label”, show the packets transmitted by the SpaceART EGSE device, whereas the right columns, below the “SpaceWire DUT label”, show the packets transmitted by the DUT. Each packet is visualized as a set of three rows following the SpW packet structure shown in Figure 1:The first row, which is yellow, shows the header of the packet, namely the logical address of the SpW packet;The second row shows the size of the packets, as the number of SpW characters;The third row is the EOP, signalling the end of the packet.

The central column, which is labelled “Time [ns]”, displays the time of transmission of the header and the EOP of each packet, with respect to the start time of the communication, expressed in nanoseconds. If needed, using a set of controls on the left of the figure, the user can visualize the time expressed in milliseconds or microseconds. The packets on both channels are aligned with respect to the time: in the case of Figure 5 we can see that the SpaceART EGSE transmits two consecutive packets, starting at time 71,699,040 and ending the transmission at time 72,710,500, then the SpaceWire DUT transmits a packet, with the header that is transmitted at time 75,301,010 and the EOP that is transmitted at time 75,415,000.This visualization allows to evaluate the timing of the communication, detecting any misbehaviour that occurred in the time-constrained scenarios. If control characters (e.g., FCT, NULL, ESC errors, TIMECODE, SYNC) are detected, the user can check a set of flags, on the left part of the GUI, to show the occurrences of such characters in the visualization. In particular, in Figure 5, FCT and SYNC values were detected, enabling the checkbox to visualize their occurrences in the table. Moreover, by clicking on a packet, the user can visualize its content at character level. Figure 6 shows the content of the second packet captured by the SpaceART Sniffer and transmitted by the SpaceWire DUT, as described by the label on the left part of the figure. In the figure, only the first 6 SpW characters of the packet are shown, but the whole content can be navigated through the GUI. The SpW characters of the packet are arranged in a table:The first column, labelled ”Id”, shows an incremental counter for the character of the packet;The second column, labelled “Time”, shows the time of transmission of the character, expressed by default in nanoseconds. Again, the user can set the visualization to express time in milliseconds or microseconds;The third column, labelled “Data”, shows the value of the character as a hexadecimal number. Using a set of controls, the user can visualize the value of each character also expressed as a decimal number or an 8 bits binary number.

Again, if any control characters were transmitted while transmitting the SpW characters composing the packet, the user can visualize their occurrences in the table, checking a set of controls above the table. In Figure 6 such controls were disabled because no control characters were transmitted during the transmission of the characters composing the visualized packet.

This visualization allows for an accurate inspection of each SpW packet, noting eventual misbehaviour in the generation and/or encapsulation of SpW data. Furthermore, the SpaceART Sniffer allows for the detection of the presence of specific SpW characters, also in communication where large quantity of data are exchanged, thanks to its filtering functionalities. The SpaceART Sniffer can exploit the SpW Packet Description Language (SpW PDL), an XML-based language defined to describe SpW packets, to match the occurrences of specific SpW packets described in [7]. In particular, the user can describe a SpW packet using the eXtensible Markup Language (XML), a human- and machine-readable markup language, defining XML files that can be given as input to the Sniffer GUI. When the Sniffer processes an XML file, it can recognize the structure of the SpW packet described in the file (i.e., the SpW characters composing the packet and its size) and detect the presence of such packet in the captured data. This process will be shown in Section 4.3, which shows the structure of an XML SpW-PDL file, and the description of its use in a test ran with the Sniffer.

Overall, the Sniffer represents a powerful instrument for SpW developers and testers who need to verify the correct functionalities of SpW-based equipment. For this reason, we have made use of the SpW Sniffer to run further tests on the Device-ICU communication presented in [5]. Such tests allowed testing the SpaceART EGSE Emulator, allowing to detect and correct eventual misbehaviour in the data generation and processing to be provided. The next chapter presents the tests that have been run and the results achieved.

## 4. Test Scenario

The test scenario presented in this paper is related to the scenario presented in [5]. In this case, the SpaceART Sniffer is used to test the communication between the SpaceART EGSE Emulator (i.e., one device) and the ICU. Figure 7 shows an overview of the test scenario.

The SpaceART EGSE Emulator is connected through SpW to the SpW0 port of the SpaceART Sniffer, whereas the ICU is connected to the SpW1 port. The Sniffer creates a transparent link between the two SpW devices, unobtrusively monitoring the connection. The ICU and the SpaceART EGSE Emulator are also connected through the SMA interface, on which the ICU transmits the SYNC signals that start each SYNC period. The use of the SpaceART Sniffer allows to validate the behaviour of the two SpW devices, ensuring that the communication constraints are matched. In particular:The SYNC period is set by the user to a custom time interval [4 ms, 120 s];The ICU sends zero or one TC packet in a single SYNC period;The SpaceART EGSE Emulator sends one or two TMs packets, depending on the operational mode set by the ICU, in a single SYNC period, carrying information about the received TC packets;The TM1 packet is transmitted within 3 ms from the reception of the SYNC signal;Both TM packets are sent, by the SpaceART EGSE Emulator, within the end of the SYNC period;

Inspecting the data captured by the Sniffer in a set of different tests, it is possible to evaluate the timing constraints (i.e., the SYNC periods) and the content of the exchanged TM and TC packets.

### 4.1. Test 1

In the first test. the SYNC period is set to 10 ms and the mode of the device is set to *safe*: only the first TM packet (i.e., TM1 packet) is transmitted by the device. On the other side, the ICU transmits 10 TC packets in 10 consecutive SYNC periods, then it stops sending TC packets and keeps receiving TM1 packets for each SYNC period. The Sniffer can be configured to trigger after many different events (e.g., the reception of a specific number of packets, the reception of a number of SYNC signals, the reception of a special character, etc.). In this case, the Sniffer was set to trigger after the transmission of 10 packets on the device channel and it was set to save data before and after the trigger event so that we could inspect the SpW data transmitted before and after the transmission of the 10th packet by the device. Figure 8 shows a snapshot of the captured data after 104,778,880 ns of communication.

The 10th packet on the left column (i.e., transmitted by the device) is highlighted with an orange header, indicating that the SpaceART Sniffer trigger was raised for this packet, as expected. In the visualization, the SYNC signals are also shown with green lines, as a signal transmitted on both channels. By evaluating the time between two consecutive SYNC signals we can notice that they are transmitted with a period of 10 ms, as set. After the arrival of the first SYNC signal (104,778,880 ns), the device transmits the TM1 packet, with header 0xDE and size 141, immediately, before the 3 ms window. In fact, the transmission of the packet starts after 37,240 ns (104,816,120) after the reception of the SYNC signal (104,778,880). After about 1 ms, the ICU transmits a TC packet. We can conclude that the time constraints resumed in Figure 3 are matched. In addition, after the 10th TC packet, in the following period, the ICU does not transmit a TC packet, as expected. This visualization of the SYNC period, provided by the Sniffer, has allowed, to refine the SpaceART EGSE Emulator, during its development phase. At first, some faults were encountered in the transmission of the Telemetry packets, with the device that was transmitting the TM1 packet without respecting the timing constraints for each period. Thanks to the use of the Sniffer we have had the opportunity to validate the device behaviour and correct it where needed, to achieve the test result shown in Figure 8.

Inspecting the characters of the TM and TC packets we can also evaluate the correctness of the content of the SpW packets generated by the device and the ICU. Even if the TM packet content details are not in the scope of this paper, in Figure 9 we can see the first part of the content of the 17th TM1 packet sent by the device.

The value of the logical address of the packet is given in the first row (id 1) and has the decimal value of 222 (0xDE), as set on the SpaceART EGSE Emulator. Other values of the TM1 packet can be inspected, for example the value with id 6 indicates the id of the last TC packet received. Its value is 10, indicating that 10 TC packets have been received. In particular, in the SYNC periods between the 10th and the 17th periods, the device has not received any TC packet, as the 10th was the last TC received, as already noticed by analysing the data traffic. Further analysis has been made on the other characters of the TM1 packet, which are described in detail in [5], as the TC Error or the Device Error fields. Such analysis allowed us to detect some minor faults in the SpW data generation provided by the SpaceART EGSE Emulator. Thanks to the use of the Sniffer, the design of the SpaceART EGSE Emulator has been refined to correct such faults, which would not have been addressed otherwise.

### 4.2. Test 2

In the second test, the SYNC period is set to 1 s and the mode of the device is set to *operational*, hence the device transmits, at least 700 µs after the reception of the SYNC signal, a TM2 packet. The TM2 packet is longer than the TM1 and its dimension depends on the data to be transmitted. In this case, being the Device emulated by the SpaceART EGSE Emulator, the TM2 packet carries a dummy content with a fixed length of 131,077 bytes.

Figure 10 shows a portion of the communication between the two SpW device, highlighting a single SYNC period. Evaluating the time of transmission of the packets we can see that the TM1 packet is transmitted before the 3 ms period and the TM2 packet is transmitted after the TM1, after 702 us with respect to the SYNC signal, as required. In this SYNC period the ICU also sends a TC packet, which transmission is interleaved with the reception of the long TM2 packet. As the TM2 packet is a long SpW packet, it requires more time to be transmitted, but its transmission ends before the end of the SYNC period, in compliance with the time constraints of the scenario. Additionally in this case, the use of the SpaceART Sniffer has allowed to detect an initial misbehaviour in the transmission of the TM2 packets: in particular, the device was transmitting TM2 packets when the device mode was set to *operational*, but, when the device mode was reset to *safe*, a last TM2 packet was wrongly transmitted. Thanks to the Sniffer we have detected the fault and corrected it in the design of the SpaceART EGSE Emulator, before running further tests on the ICU.

### 4.3. Test 3

In the third test, the XML filter of the Sniffer is exploited to analyse the data captured in Test 1. An XML packet description of the TM and TC packets has been defined, in compliance with the SpW PDL defined in [7]. The database is filtered to match the occurrences of these packets in the communication. This way the user can evaluate a larger portion of data, not only by visualizing the data through the GUI, but also by receiving statistics about the communication just by defining a human-readable XML file. This scenario covers a 10 s test with one thousand SYNC periods of 10 ms. Even if the user can check the timing of the packets by the GUI visualization (Figure 8), it would be unfeasible to analyse the actual transmission of all the expected 1000 TM packets and evaluate their content. The SpW PDL file can be used to filter data in the database [7]: the Software operates a comparison between all the found packets and the XML definition, checking if the packet transmitted by the SpW device match the XML definition. In particular, a SpW PDL for each of the three packets involved in the communication has been defined (i.e., TM1, TM2, and TC packets). Figure 11 shows the XML definition of the TM1 and TC packets, in compliance with SpW PDL, hiding their whole content for the sake of simplicity.

Each packet description is encapsulated in a “spw_packet” tag with a “name” attribute reporting the name of the packet (TM1, TC, TM2), used to label the detected packet in the GUI. For each packet the expected value of the logical address is defined in the “address” tag: for example, for TM1 packet the logical address admits one of the three indicated values (222, 238, 254). Such values will be inspected by the Sniffer GUI to detect the packets that have the specified logical address. The “payload” tag, not shown here, contains the description of each expected field of each packet, along with their size. Moreover for each field of the payload the user can define a default value for the filtering packet, for example to filter each TM1 packet in which a specific character has a specific value (i.e., TC error character = 1). The use of the SpW PDL allows to quickly investigate the content of the transmitted SpW packets, detecting the presence of the defined packet, in this case the TM and TC packets and finding eventual unexpected transmitted SpW data.

Figure 12 shows the found packets and the statistics provided by the GUI after filtering the data with the SpW PDL, in a visualization that is similar to the one in Figure 5. At the bottom of the window, the statistics about the exchanged packets is shown, highlighting the number of packets of a specific type that have been detected, among all the SpW packets. As expected, 1000 TM1 packets were transmitted by the device and only 10 TCs were transmitted by the ICU. The XML visualization of the data allows to see the occurrences of each packet. In the case of Test 2, we would have also found occurrences of TM2 packets on the device table. Thanks to the XML definition we can also inspect the content of the found packets, checking the content of each TM packet in a user-friendly visualization, matching the fields defined in the payload field.

## 5. Results

The SpaceART SpW Sniffer has been successfully employed, along with the SpaceART EGSE Emulator described in [5], to comprehensively validate the described ICU. As shown in [5], the SpaceART EGSE Emulator has been employed to run a set of tests, to cover many corner-cases and verify the compliance of the DUT with the constraints of a given scenario. The set of tests described in this paper has provided further information to validate the ICU, thanks to the adoption of the SpaceART SpW Sniffer EGSE.

A critical point in the validation is the compliance with the given time constraints, which has been successfully tested thanks to the use of the SpaceART SpW Sniffer. The unobtrusive monitoring of the link has allowed us to validate the communication from an external point of view, accurately measuring the occurrences of SYNC periods and the transmission time of every packet involved in the SpW communication.

The first test allowed us to verify the synchronization between the SpaceART EGSE Emulator (i.e., the device) and the ICU. Thanks to the Sniffer GUI, the SYNC period duration has been checked and the correct transmission of TM1 and TC packets has been proved, also analysing their contents at a character level.

The second test allowed us to verify the correct transmission of TM2 packets by the device. The use of the Sniffer has shown compliance with the timing constraints of the scenario: hence, the long TM2 packets were transmitted at least 700 µs after the reception of a SYNC signal, and within the reception of the consecutive SYNC signal.

Moreover, the third test employed the SpW PDL filter of the Sniffer, to verify the absence of spurious packets and the occurrences of TM1 and TC packets in a large communication, in which it could be infeasible for the user to verify each packet, one by one.

Finally, the SpaceART SpW Sniffer and the SpaceART EGSE Emulator allowed us to efficiently cover the DUT verification. The combined use of these EGSEs has provided results that could have not been accomplished using other general-purpose state-of-the-art SpW EGSE. Given the constraints and the specificity of the test scenario, the use of the SpaceART EGSE Emulator, instead of a general purpose EGSE, has allowed us to:Generate mission-related packets, which content is used to stimulate definite behaviours of the DUT (i.e., the ICU);Transmit SpW packets in specific time-slots, as required by the DUT nominal working process;Process the DUT telecommand with the deserved accuracy and transmit the required telemetry packet.

The use of a general-purpose SpW EGSE would not have provided us with an efficient testing of all the possible corner-cases involved in the Device-ICU communication, thus denying an efficient verification of the compliance with the depicted time constraints. Moreover, the use of the SpaceART Sniffer for link analysis, has allowed us to inspect the data communication from an external point of view, helping in the development and verification of the SpW EGSE Emulator and in the testing of the Device-ICU communication. Again, this contribution has been crucial for the test of the ICU and would not have been possible without the use of the SpaceART Sniffer. This approach has provided a comprehensive testing of the ICU, contributing to the verification of the SpW on-board communication for the specific mission.

## 6. Conclusions

In this paper, the use of the SpaceART SpW Sniffer and the SpaceART EGSE Emulator to run a comprehensive set of tests on a mission-related SpW communication has been presented. The tests shown in this paper extend the validation tests described in [5]. The combined use of the SpaceART SpW Sniffer and the SpaceART EGSE Emulator has allowed us to verify the correct functionalities of the on-board SpW communication between a SpW-based device and an ICU. The customization of the SpaceART EGSE Emulator has allowed us to stimulate the corner-case behaviour of the ICU and, at the same time, the use of the SpaceART SpW Sniffer for link analysis, and allowed us to inspect the communication from an external point of view, providing a comprehensive analysis of data traffic and timing.

First, an introduction of the SpaceWire standard [3] and the state-of-the-art of SpaceWire EGSE and Link Analysers [6,8,10,12,14,15,16,17] was given. Then, an overview about the SpaceART EGSE Emulator, described in [5], and the TM and TC commands [19] generated and transmitted in the test case were provided. An overview about the SpaceART SpW Sniffer, a SpaceWire link analyser [6] was given, in order to show its functionalities and its importance for testing SpW-based communication. Finally, the set of tests run using the SpaceART SpW Sniffer were presented. These tests allowed us to verify the SpW communication between the SpaceART EGSE Emulator and the ICU, in a scenario in which specific SpW packets with definite time constraints must be exchanged. The use of the SpaceART EGSE Emulator allowed us to efficiently test the ICU, covering every possible test-scenario. On the other side, the use of the SpaceART SpW Sniffer for unobtrusively monitoring the link, has given us further information on the structure and the timing of the exchanged packets from an external point-of-view.

## Figures and Tables

**Figure 1 sensors-23-01580-f001:**
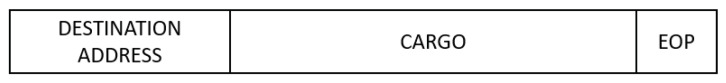
SpaceWire (SpW) packet structure.

**Figure 2 sensors-23-01580-f002:**
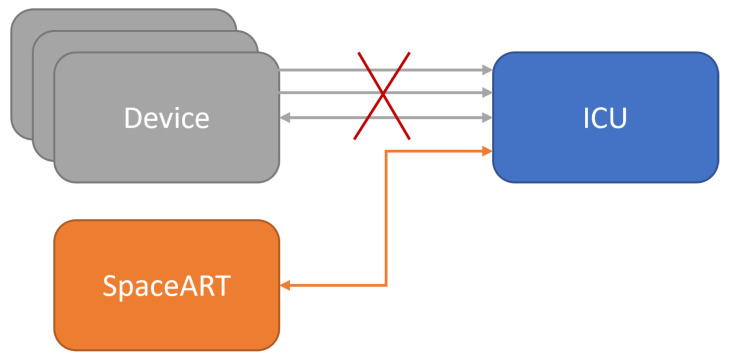
SpaceART EGSE emulator-Instrument Control Unit (ICU) test scenario.

**Figure 3 sensors-23-01580-f003:**
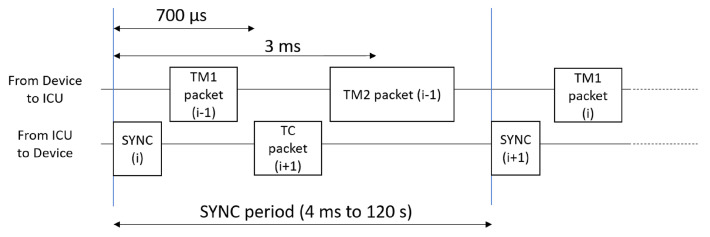
Time diagram of ICU-device communication.

**Figure 4 sensors-23-01580-f004:**
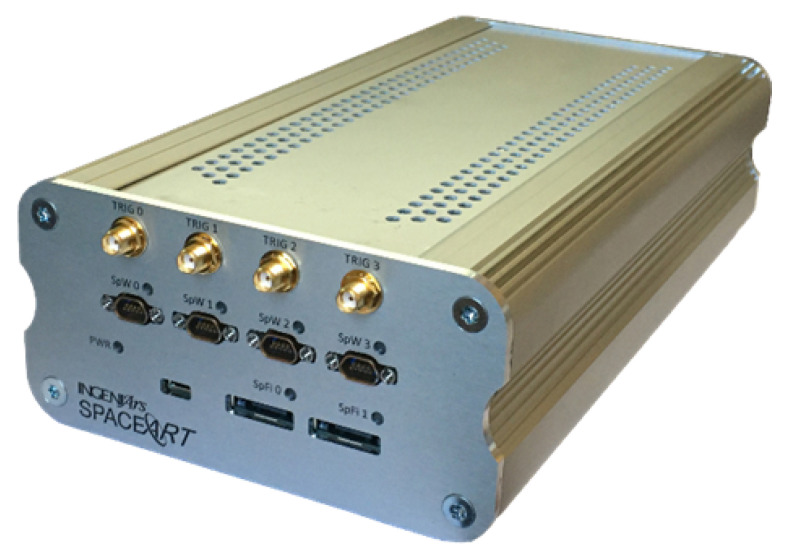
IngeniArs SpaceART Electrical Ground Support Equipment (EGSE) hardware unit.

**Figure 5 sensors-23-01580-f005:**
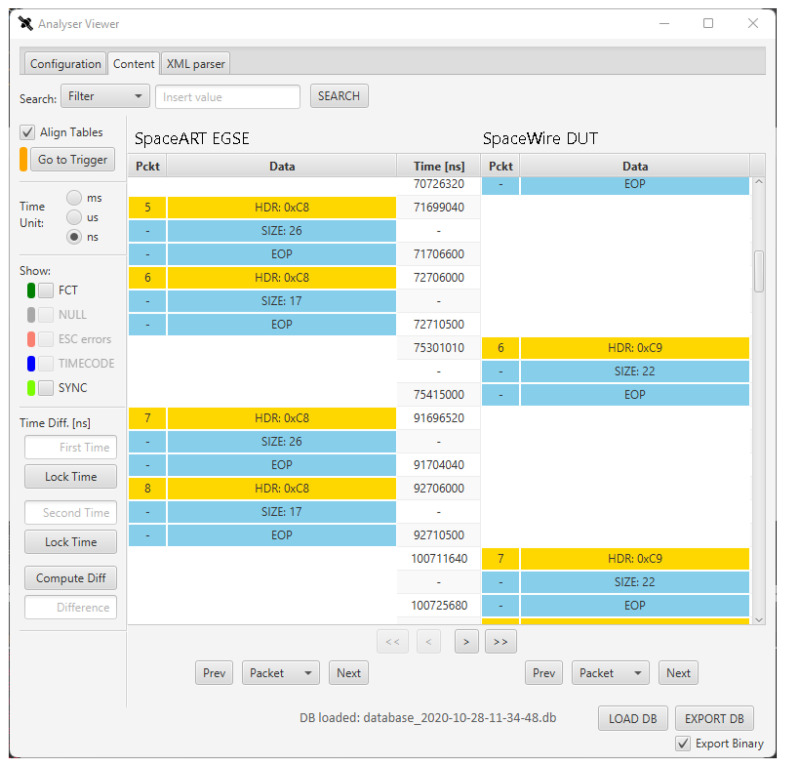
SpaceART Sniffer Graphical User Interface (GUI) visualization of the packet exchanged between a SpaceART EGSE and a SpaceWire Device Under Test (DUT).

**Figure 6 sensors-23-01580-f006:**
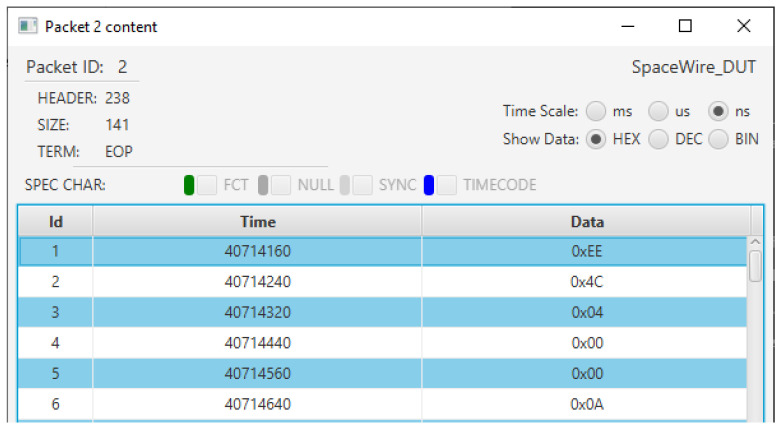
SpaceART Sniffer GUI partial visualization of the content of the second SpW packet transmitted by the SpaceWire DUT.

**Figure 7 sensors-23-01580-f007:**
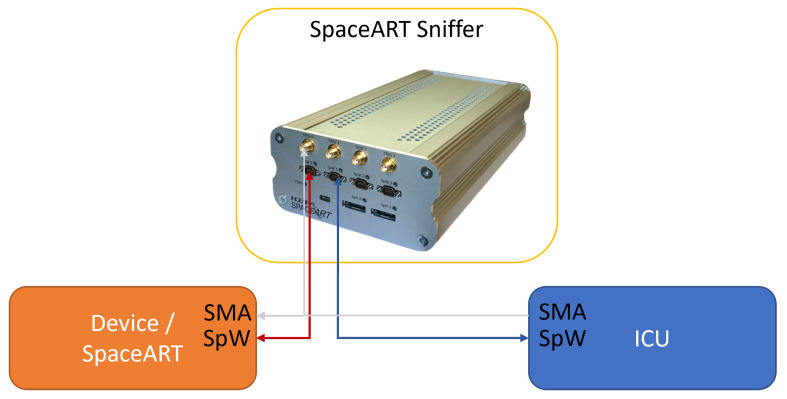
SpaceART SpW Sniffer is used to unobtrusively monitor the SpW connection between the emulated device and the ICU.

**Figure 8 sensors-23-01580-f008:**
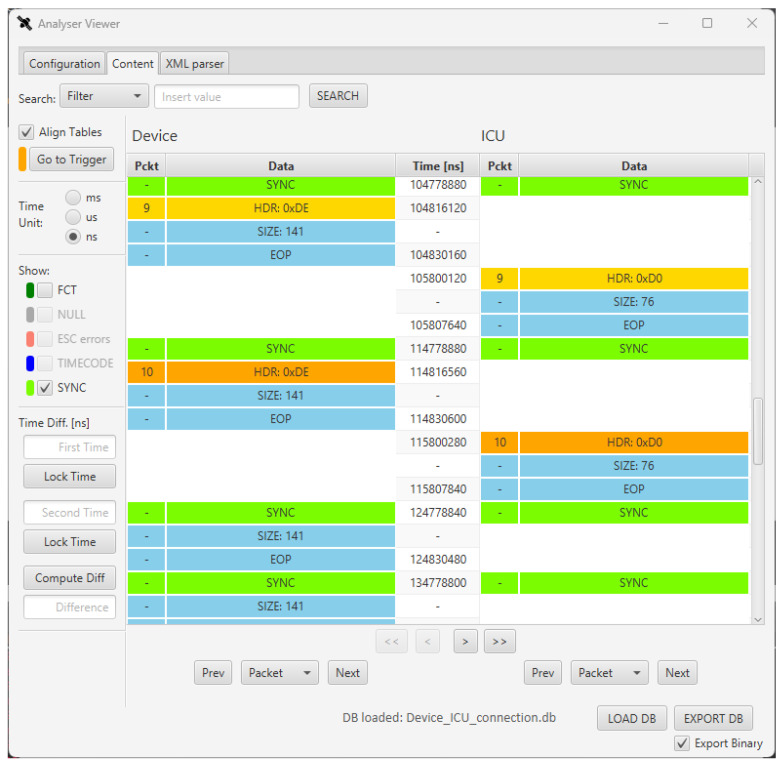
SpaceART Sniffer GUI visualization of packet exchanged in Test 1.

**Figure 9 sensors-23-01580-f009:**
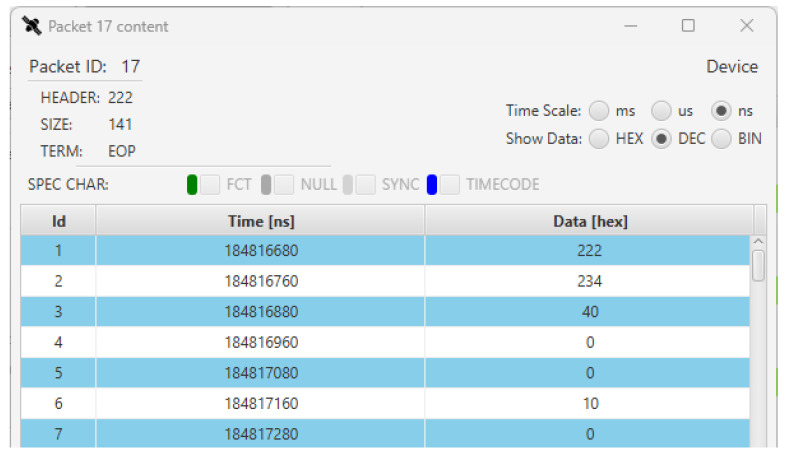
SpaceART Sniffer GUI visualization of the content of the 17th TM1 packet transmitted by the device in Test 1.

**Figure 10 sensors-23-01580-f010:**
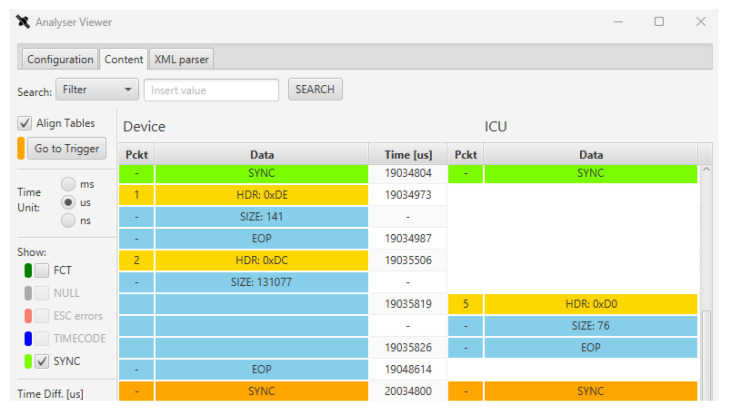
SpaceART Sniffer GUI partial visualization of TM1 and TM2 packets transmitted by the device.

**Figure 11 sensors-23-01580-f011:**
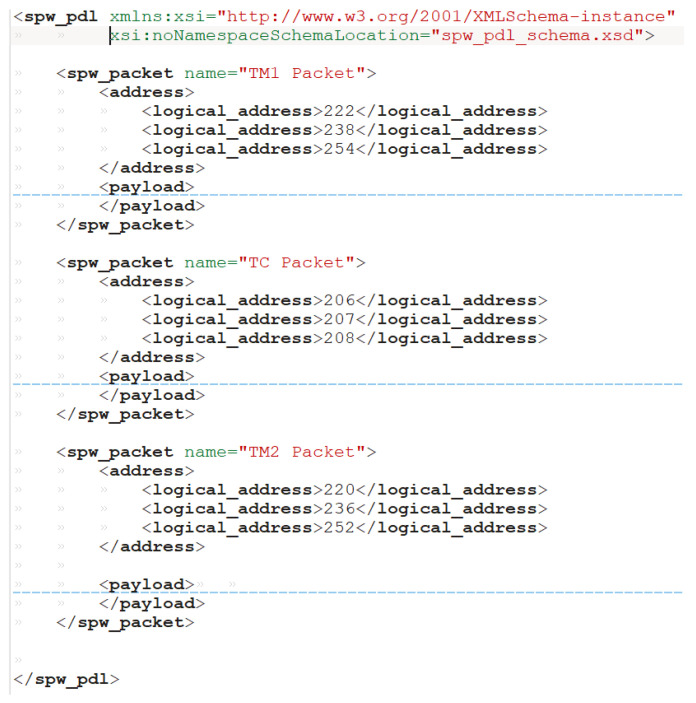
XML packet description of TM1, TM2, and TC packets overview, in compliance with SpW Packet Description Language (PDL) [7].

**Figure 12 sensors-23-01580-f012:**
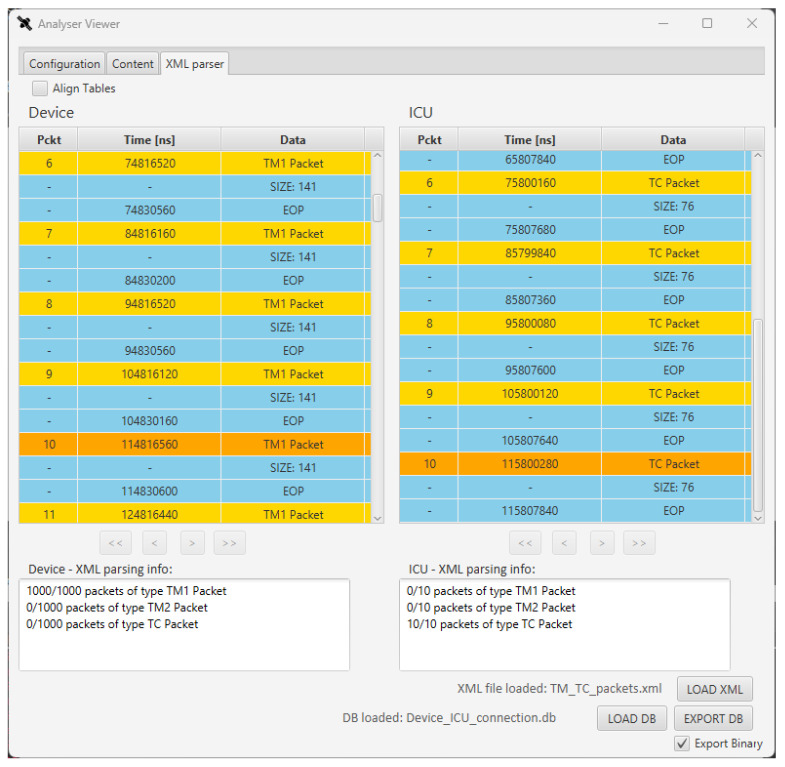
SpaceART Sniffer GUI visualization of packets filtered by means of SpW PDL.

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
