# Peer review of "SpaceART SpaceWire Sniffer for Link Monitoring: A Complete Communication Analysis in a Time-Constrained Test Scenarioâ€"

_sensors, 2023, doi:10.3390/s23031580_

Round 1
Reviewer 1 Report
In this paper, the problem of spacewire sniffer for link monitoring is investigated. It is interesting. The main contributions of the paper are major. It is acceptable for publication.
Author Response
On behalf of all the authors, I would like to thank you for taking the necessary time and effort to review the manuscript. Moreover, I am honored that you appreciate the manuscript and you consider it acceptable for publication.
Kind regards,
Roberto Ciardi
Reviewer 2 Report
I would be interesting to mention if you detect any faults/misbehaviour during the test and if there is any fault diagnosis performed during test runs ?
Reviewer 3 Report
1. It is suggested to refine the innovation points of the paper, and give the improvements compared with similar technologies.
2. Please appropriately add references for technical comparison.
Author Response
On behalf of all authors, I would like to thank you for your precious comment.
Please see the attachment.

Reviewer 4 Report
I can recommend the publication of this manuscript. The authors have demonstrated that the SpaceART SpW Sniffer can provide unobstructed monitoring of the exchanged data in a SpW network. The tests are comprehensive.
Author Response
On behalf of all the authors, I would like to thank you for taking the necessary time and effort to review the manuscript. Moreover, I am honored that you appreciate the manuscript and you recommend its publication.
Kind regards,
Roberto Ciardi
Reviewer 5 Report
1- the paper is written in a way that is difficult to understand.
2- The paper contribution is very limitted and not explained properly.
3- The limitations of SpaceART SpaceWire Sniffer should clearly be discussed.
4- The figures not clear,
5- The abstract and conclusion sections need to be improved.
1 6- The Introduction section should provide more insight into the topic.
2
Author Response
On behalf of all authors I would like to thank you for your precious comments.
Please see the attachment.

Round 2
Reviewer 5 Report
The paper can be accepted in its current form.